# An Approach to Improve the Effectiveness of the Video-Assisted Intubating Stylet Technique for Tracheal Intubation: A Case Series Report

**DOI:** 10.3390/healthcare11060891

**Published:** 2023-03-20

**Authors:** Cing-Hong Lan, Hsiang-Ning Luk, Jason Zhensheng Qu, Alan Shikani

**Affiliations:** 1Department of Anesthesia, Hualien Tzuchi Hospital, Hualien 97002, Taiwan; 2Bio-Math Laboratory, Department of Financial Engineering, Providence University, Taichung 43301, Taiwan; 3Department of Anesthesia, Critical Care and Pain Medicine, Massachusetts General Hospital, Harvard Medical School, Boston, MA 02114, USA; 4Division of Otolaryngology-Head and Neck Surgery, LifeBridge Sinai Hospital, Baltimore, MD 21218, USA; 5Division of Otolaryngology-Head and Neck Surgery, MedStar Union Memorial Hospital, Baltimore, MD 21218, USA

**Keywords:** airway management, difficult airway, tracheal intubation, laryngoscopy, videolaryngoscopy, video-assisted intubating stylet, anesthesia, nasopharyngeal airway, suction catheter

## Abstract

Direct laryngoscopy and videolaryngoscopy are currently the dominant tools for endotracheal intubation. However, the video-assisted intubating stylet, a type of videolaryngoscopy, has been shown to offer some advantages over these tools, such as rapid intubation time, high first-attempt success rates, less airway stimulation, and high subjective satisfaction. On the other hand, this optical intubating technique also has some technical limitations that need to be addressed, including camera lens fogging, airway path disorientation, and obscured visibility due to secretions. In this clinical report, we describe an approach that improves the visibility of the glottis by introducing a suctioning catheter into the nasopharyngeal airway to enhance the efficiency and accuracy of using the intubating stylet technique for tracheal intubation.

## 1. Introduction

Direct laryngoscopy (DL) had long been recognized and maintained as the holy grail of intubating tools until videolaryngoscopy (VL) was introduced two decades ago [1,2]. The unique design of the optic device element that is incorporated into the VL technique has improved the visualization and first-attempt success rate of tracheal intubation compared to the performance of DL [3]. Some other advantages of VL over DL are the short intubation time, less sympathetic overstimulation, and fewer airway injuries [4,5,6]. 

Recently, an illuminating and visualizing intubating stylet has been designed as an alternative tool for tracheal intubation [7]. In addition to the Lightwand [8], a new “seeing” stylet scope was invented in 1999 [9]. Whereas intubation using traditional stylets is a blind or semi-blind technique, “seeing” stylets allow for continuous monitoring of the hypopharynx and airway during the introduction of the tube into the glottis [9]. The wide clinical application of this video-assisted intubating stylet technique (VS; also known as the Shikani technique) has since been reported in the literature (for a review, see [10]). 

Following the original VS design (Clarus Shikani, CLARUS MEDICAL, LLC, Minneapolis, MN 55441, USA), more than a dozen similar products have become commercially available. The use of VS has been described in several clinical scenarios [11,12,13,14]. The VS technique has been widely used in both routine and emergency airway management (on approximately 7000 patients per year in our medical center since 2016), which has been documented in recent clinical reports [15,16,17,18,19,20,21,22,23]. In most cases, the optic design provides a better visualization of the upper airway pathway (Figure 1A,B) and makes the tracheal intubation much easier, smoother, quicker, and less traumatic (for a review, see [10]). 

Similar to the Bonfils intubation endoscope and fiberoptic scopes [24,25], the VS technique can be encumbered by copious secretions, blood, or vomitus in the airway (Figure 1C,D). These secretions and mucus have to be carefully removed via suctioning to improve the visualization of the airway anatomy. Additionally, the lens of the VS device may need to be cleaned. Another difficulty with the VS technique is the loss of orientation or course of direction, which can be caused by the airway structures of patients.

In this clinical report, we share our experiences of using a simple modified suction system (Figure 2A) while applying the VS technique for tracheal intubation (Figure 2B). Briefly, the suction device was composed of a flexible suction catheter and a soft nasopharyngeal airway with a blunt tip. The catheter was inserted into the airway and the tip of the catheter was left sticking out of the airway by approximately 2 cm (Figure 2A). There were three holes in the catheter tip for suction purposes. In order to maintain the constant position of the catheter tip, the catheter was firmly taped onto the proximal end of the airway. The depth of the insertion of the catheter–airway assembly into the patient’s oropharyngeal space was estimated according to the distance between the patient’s lip and the thyroid cartilage. An adjustable flange was placed on the airway, which assured that the depth was appropriate for each individual patient. 

## 2. Case Series Presentation

In this report, the following VS devices were used: (1) C-MAC-VS (video stylet), KARL STORZ SE & Co., Ltd. KG, Tuttlingen, Germany; (2) TuoRen Kingtaek video intubating stylet, TuoRen, Henan TuoRen Medical Device Co., Ltd. Xinxiang, Henan, 453401, China; and (3) Trachway video intubating stylet, Markstein Sichtech Medical Corp., Taichung, 407, Taiwan. The nasopharyngeal airways (i.e., the Wirupren^®^ adjustable flanges and Rusch^®^ PVC nasal airways) were from Teleflex^®^ Medical, IDA Business and Technology Park, Dublin Road, Athlone, Co., Ltd. Westmeath, Ireland. In all cases, standard American Society of Anesthesiologists (ASA) monitoring (including ECG, NIBP, SpO_2_, ToF, and ETCO_2_.) was applied intraoperatively. The induction and maintenance of anesthesia followed the routine protocols, including the use of medications such as midazolam, fentanyl, propofol, neuromuscular blocking agents, and inhalational anesthetics. Other relevant information was summarized in Table 1. All the tracheal intubations were performed using the VS technique. The catheter–nasopharyngeal airway assembly (Figure 2) was gently inserted into the patient’s oropharyngeal space to eliminate any possible secretions. This airway also served as a visual guide for positioning the stylet in front of the glottic opening.

### 2.1. Case 1 (Limited Cervical Spine Mobility Due to Neck Collar)

A 56-year-old man (height 160 cm; weight 60 kg) suffered an intervertebral disc herniation following trauma (HIVD and C3-7). He underwent a laminoplasty (C3-6) and an anterior cervical discectomy and fusion (ACDF, C3-4, and C6-7). The range of motion in the patient’s cervical spine was limited by a cervical collar (Figure 3A, upper panel). The suction catheter passed underneath the epiglottis easily, so there was no need to lift the epiglottis, which minimized the movement of the neck (middle panel). The glottis could be seen clearly (lower panel). The intubation time (from lip to trachea) was 58 s.

### 2.2. Case 2 (Limited Cervical Spine Mobility Due to Headframe)

A 58-year-old man (height 158 cm; weight 91 kg) was admitted for Leksell coordinate frame-based stereotactic neurosurgery (Figure 3B). Bilateral subthalamic nucleus (STN) deep brain stimulation (DBS) using microelectrode recording (MER) and an implantable pulse generator (IPG) was performed under general anesthesia. A stereotactic head frame was secured on a Mayfield adaptor before the induction of anesthesia (Figure 3B, upper panel). The intubating time (from lip to trachea) was 21 s.

### 2.3. Case 3 (Morbid Obesity)

A 25-year-old man (height 183 cm; weight 150 kg) was admitted for exotropia correction (i.e., bilateral rectus recession and inferior oblique muscle myectomy) under general anesthesia. The patient was diagnosed with morbid obesity and obstructive sleep apnea syndrome (Figure 3C). The patient was preoxygenated in the ramp position. The intubating time (from lip to trachea) was 20 s.

### 2.4. Case 4 (Morbid Obesity with Enlarged Tonsils)

A 27-year-old woman (height 153 cm; weight 99 kg) was admitted for the transsphenoidal removal of a pituitary cyst under general anesthesia. The enlarged tonsils (grade 3) blocked the pharyngeal space (Figure 4A, upper panel). The intubating time (from lip to trachea) was 42 s.

### 2.5. Case 5 (Enlarged Thyroid Glands)

A 47-year-old man (height 176 cm; weight 74 kg) was admitted for a thyroidectomy under general anesthesia. The patient was diagnosed with a multinodular goiter of the bilateral thyroid glands. An enlarged thyroid gland (11.8 cm × 4.3 cm in size with central necrosis) was observed in the right thyroid lobe (Figure 4B, upper panel). A total thyroidectomy (right lobe) and a partial thyroidectomy (left lobe) were scheduled. The intubating time (from lip to trachea) was 15 s.

### 2.6. Case 6 (Oral Cancer)

A 63-year-old woman (height 145 cm; weight 54 kg) was admitted for a gum tumor wide excision, a partial maxillectomy, a complex extraction, and a tracheostomy under general anesthesia. The patient was diagnosed with an upper gum squamous cell carcinoma (pT2N0M0, perineural invasion (–), and lymphovascular invasion (–)) (Figure 4C). A neuromuscular blockade was induced using succinylcholine and rocuronium for the tracheal intubation. A priming dose of rocuronium, followed by a relaxation dose of succinylcholine, were applied to lessen muscular fasciculations and to enhance muscle relaxation for the tracheal intubation. The intubating time (from lip to trachea) was 23 s.

### 2.7. Case 7 (Facial Trauma)

A 45-year-old man (height 168 cm; weight 78 kg) was admitted for an open reduction, a nerve graft, and a deep complicated facial wound debridement under general anesthesia (Figure 5A). The patient was diagnosed with deep facial lacerations, a nasal bone fracture, and injuries to the left trigeminal nerve due to trauma from a car accident. The patient was preoxygenated with high-flow O_2_ via face mask ventilation. The intubating time (from lip to trachea) was 20 s.

### 2.8. Case 8 (Deep Neck Infection)

A 68-year-old woman (height 158 cm; weight 77 kg) was admitted for the incision and open surgical drainage of the neck region and a tracheostomy under general anesthesia. The patient was diagnosed with a deep neck infection (left parapharyngeal space abscess, Figure 5B). The patient was preoxygenated with high-flow O_2_ via face mask ventilation. A neuromuscular blockade was induced using succinylcholine for the tracheal intubation. The intubating time (from lip to trachea) was 19 s.

### 2.9. Case 9 (Rapid Sequence Induction and Intubation)

An 18-year-old woman (height 160 cm; weight 60 kg) was admitted for laparoscopic appendectomy under general anesthesia. The patient was diagnosed with acute appendicitis. The patient was preoxygenated with high-flow O_2_ via face mask ventilation in the ramp position (Figure 5C). General anesthesia for the rapid sequence induction and intubation (RSII) was induced via intravenous lidocaine, fentanyl, and propofol. A neuromuscular blockade was induced using rocuronium and succinylcholine for the tracheal intubation. The cricoid pressure (CP) maneuver was released immediately before the tracheal intubation started in order to avoid the distortion and compression of the glottis by the maneuver. The role of CP for RSII remains controversial, and we should be cautious of regurgitation caused by the sudden release of CP. The intubating time (from lip to trachea) was 9 s.

## 3. Discussion

In this case series report, we presented nine cases in which the VS technique was performed using a modified suction catheter–nasopharyngeal airway assembly for tracheal intubation (Table 1). Using this method, tracheal intubation could be accomplished smoothly, swiftly, and accurately. This was supported by several observed parameters, such as glottis visibility, first-attempt success rate, intubation time (from lip to trachea), complications (dental/soft tissue injuries, hypoxemia, hypotension, autonomic stimulation, etc.), and the subjective satisfaction of the airway operator (Table 1). Airway management has evolved over the last few decades to accommodate a variety of challenges [26,27]. Several new tools and techniques have been developed to improve and facilitate both routine and emergency endotracheal intubation (e.g., VL, laryngeal masks, optic stylets, etc.). New practical clinical guidelines and algorithms have also been developed to improve safety and minimize adverse effects [28,29,30,31]. The conventional role of DL and the evolutionary role of VL have been widely studied and discussed. The VS technique has also been extensively described and reported [10].

Since Shikani invented the “seeing” stylet scope and described the VS technique in 1999 [9], there has been a paradigm shift in airway management, particularly for difficult airways. The VS technique has been tested and trialed in many clinical scenarios, such as limited cervical spine mobility, morbid obesity, etc. [32,33]. When patients are in the supine position, the position-dependent drooping of the epiglottis can completely block the view of the glottis, constituting an insurmountable problem for tracheal intubation. To solve this problem, several practical solutions have been developed for VS instruments. For example, the elevation of the epiglottis could be achieved via an effective jaw thrust maneuver (by the airway operator or the second airway assistant). Although the jaw thrust maneuver can be helpful in elevating the epiglottis to some extent (e.g., 1 cm), it could be insufficient due to patient factors (e.g., limited cervical spine mobility, morbid obesity, ankylosing spondylitis, etc.). If the jaw thrust maneuver is not possible, combined use of tongue elevators or DL/VL can be used to lift the epiglottis by introducing the blade tip into the vallecular region. It is worth mentioning that the combined use of the VS technique and VL/DL could be useful in patients with a difficult airway (e.g., caused by copious secretions in their airways, a large epiglottis, or a downfolding of the epiglottis) [23,34,35,36]. Direct high-flow oxygenation has been shown to remove secretions and improve the visualization of the airway during application of the VS technique [37]. For emergency or significant vomitus or secretions, the SALAD technique (suction-assisted laryngoscopy for airway decontamination) can be helpful [38,39,40]. During the regular induction of anesthesia and tracheal intubation processes, both regular flexible suction catheters and rigid Yankauer suction tips are commonly used to remove any substances that could impede the visibility of the airway. In contrast with the SALAD technique, our soft suction-guide assembly could also achieve the goal of clearing the airway before applying the VS technique, which is particularly crucial for novice practitioners and students.

The VS technique was tested with both experienced and novice practitioners on mannequin models, and the performance and learning curve of applying the VS technique were found to be acceptable (i.e., intubation time, first-attempt success rate, subjective satisfaction, etc.) [41,42,43,44,45,46,47]. However, during the routine application of the VS technique, several factors (e.g., human, patient, and mechanical factors) can affect its success. The most common challenge for the technique is poor visibility caused by soft tissue obstructions, tissue contact, and secretions (e.g., Figure 1). Therefore, using our suction guide assembly (Figure 3, Figure 4 and Figure 5) could enable the airway operator to adequately eliminate secretions, easily orient and find the way to the glottis, and create enough space below the epiglottis to allow for the smooth and swift passage of the stylet–endotracheal unit.

## 4. Conclusions

The main limitations of this case series report include (1) the retrospective study design; (2) the limited possibility of generalizing the validity and reliability of the study due to the small number of subjects; and (3) the fact that the study was conducted by a single airway operator in a single medical center. Therefore, the use and interpretation of this case series report should be carefully considered within a much broader context in order to avoid the unnecessary overstatement of the validity of the study. Although DL and VL still play important roles in airway management in many countries and regions, the emergence of the new VS technique has gradually prevailed over recent years. This technique has been demonstrated to be effective in both routine airway management and certain difficult clinical scenarios, including head or neck tumors, morbid obesity, cervical spine immobility, rapid sequence intubation, etc. [10]. However, according to the current guidelines, it should be mentioned that when a difficult airway (e.g., difficult laryngoscopy, face mask ventilation, emergency invasive airway, etc.) is anticipated/suspected, the awake intubation option should be seriously considered [31]. In conclusion, our simple nasopharyngeal tube–suction catheter assembly could improve the efficiency and effectiveness of the video-assisted intubating stylet technique for both experienced practitioners and novice trainees.

## Figures and Tables

**Figure 1 healthcare-11-00891-f001:**
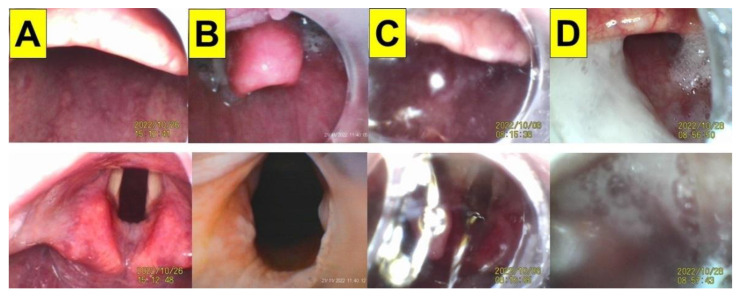
The airway scenarios during the routine application of the video-assisted intubating stylet technique in four patients. Upper panels: the view above the epiglottis; lower panels: the full glottis visualization. (**A**) A 46-year-old man with a body mass index (BMI) of 26.8 kg/m^2^ (height 181 cm; weight 88 kg). (**B**) An 85-year-old man with a BMI of 26.7 kg/m^2^ (height 163 cm; weight 71 kg). The intubation time using the intubating stylet (from lip to trachea) was 30 s in (**A**) and 25 s in (**B**). (**C**,**D**) Two examples of the interference from copious amounts of secretions, mucus, and thick sputum on the visibility of airways during the VS technique. (**C**) A 45-year-old woman with a BMI of 24.2 kg/m^2^ (height 160 cm; weight 62 kg); (**D**) a 51-year-old man with a BMI of 26.4 kg/m^2^ (height 177 cm; weight 83 kg). The intubation time was 55 s in (**C**) and 85 s in (**D**) (see Appendix A). The intubating devices used were TuoRen Kingtaeks and a Trachway.

**Figure 2 healthcare-11-00891-f002:**
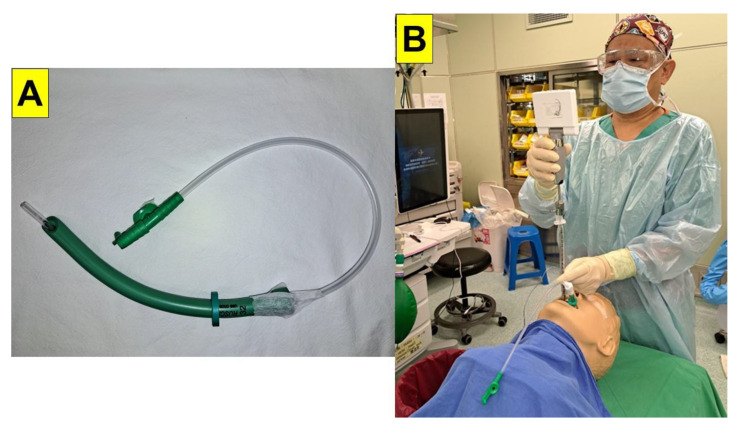
The suction catheter–nasopharyngeal airway assembly: (**A**) The suction catheter was inserted into a nasopharyngeal airway and the tip of the catheter (approximately 2 cm in length) was left sticking out of the distal end of the airway. The catheter was taped firmly to the proximal end of the airway. The distance between the flange and the tip of the suction catheter was determined by estimating the distance from the patient’s lip to their thyroid cartilage. (**B**) The combined use of the VS technique and the suction–guide assembly on a mannequin model (S-RVL video stylet, Sensorendo Medical Technology, Zhuhai, China).

**Figure 3 healthcare-11-00891-f003:**
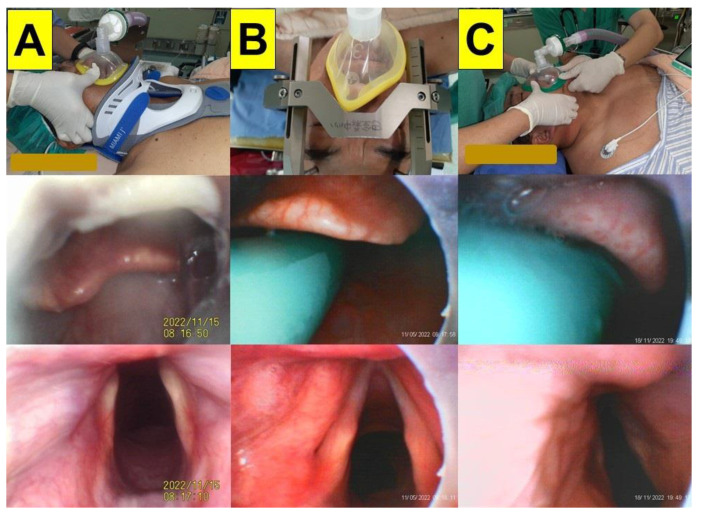
The application of the VS technique using a suction catheter–nasopharyngeal airway assembly in case 1 (**A**), case 2 (**B**), and case 3 (**C**). The upper panel: during facemask ventilation; the middle panel: the view above the epiglottis; the lower panel: the full glottis visualization. (See Appendix A).

**Figure 4 healthcare-11-00891-f004:**
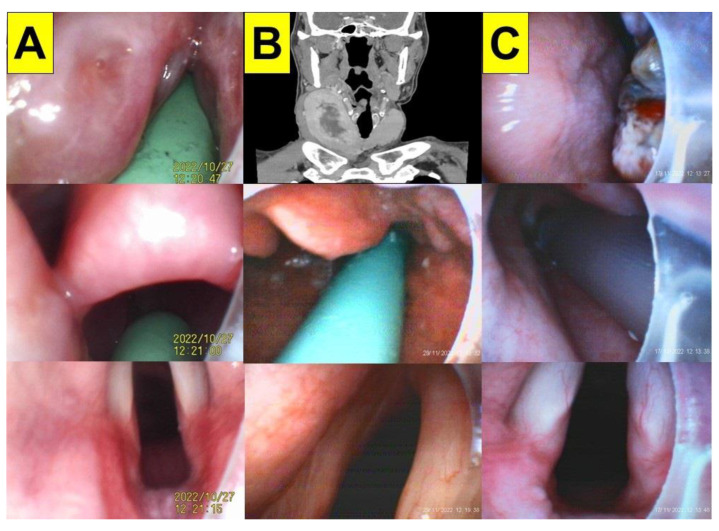
The application of the VS technique using a suction catheter–nasopharyngeal airway assembly in case 4 (**A**), case 5 (**B**), and case 6 (**C**). The upper panel: enlarged tonsils (**A**), goiter (**B**), and gum cancers (**C**); the middle panel: the view above the epiglottis; the lower panel: the full glottis visualization. (See Appendix A).

**Figure 5 healthcare-11-00891-f005:**
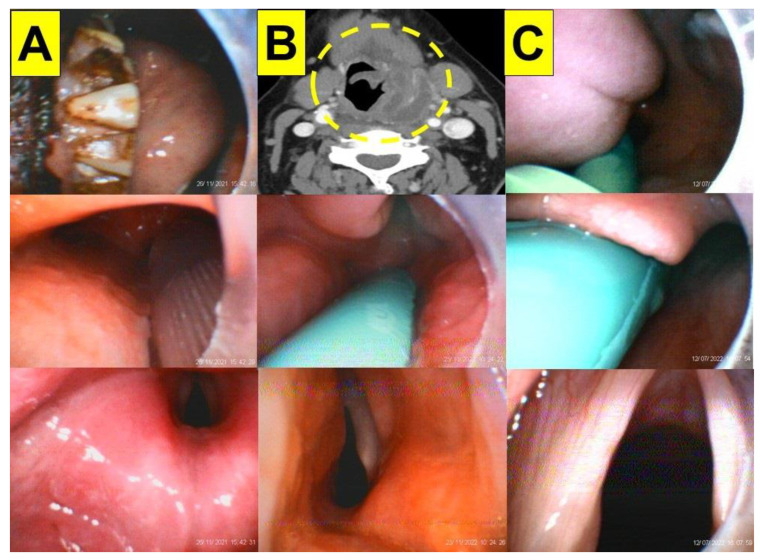
The application of the VS technique using a suction catheter–nasopharyngeal airway assembly in the case 7 (**A**), case 8 (**B**), and case 9 (**C**). The upper panel: (**A**) facial trauma; (**B**) deep neck infection (“rim enhancement sign”) of a surgically drainable abscess (yellow circle) on contrast-enhanced CT scans; and (**C**) rapid sequence induction and intubation; the middle panel: the view above the epiglottis; the lower panel: the full glottis visualization. (See Appendix A).

**Table 1 healthcare-11-00891-t001:** A summary of the nine cases in this series report.

	Case 1	Case 2	Case 3	Case 4	Case 5	Case 6	Case 7	Case 8	Case 9
Age/Gender	56/M	58/M	25/M	27/F	47/M	63/F	45/M	68/F	18/F
BMI (kg/m^2^)	23.4	36.4	44.7	42.2	23.8	25.6	27.6	30.8	23.4
ASA Physical Status	II	III	II	III	I	I	I	III	II
Comorbidity	Hypertension	Hypertension, type II diabetes mellitus, and gout	Morbid obesity	Exotropia, benign intracranial hypertension, Foster Kennedy syndrome, morbid obesity, and obstructive sleep apnea syndrome.				Type II diabetes mellitus, hypertension, gall stones, lumbar spine disc herniation, gastroesophageal reflux disease, and esophagitis.	Systemic lupus erythematosus
Diagnosis	Cervical HIVD	Parkinson’s disease	Exotropia	Pituitary cyst	Thyroid goiter	Gum tumor	Facial trauma	Deep neck infection	Acute appendicitis
Surgery	ACDF	DBS	Recession and myectomy	Transsphenoidal removal	Thyroidectomy	Wide excision maxillectomy	Open reduction, graft, and wound debridement	Incision, drainage, and tracheostomy	Laparoscopic appendectomy
Induction	MDZ, FEN, PPF, and ROC	FEN, PPF, and ROC	FEN, PPF, and ROC	MDZ, FEN, PPF, and ROC	FEN, PPF, and ROC	FEN, PPF, ROC, and SCC	FEN, PPF, ROC, and SCC	MDZ, FEN, PPF, and SCC	FEN, PPF, ROC, and SCC
Maintenance	Desflurane	Sevoflurane	Sevoflurane	Sevoflurane	Sevoflurane	Sevoflurane	Sevoflurane	Sevoflurane	Sevoflurane
Airway Evaluation	Cervical collar	Stereotactic frame	Thick neck	Enlarged tonsils	Neck mass	Gum tumor mass	Blood in the oral cavity	Neck erythema, swelling	Normal
LQS Grading	Grade 2	Grade 2	Grade 2	Grade 2	Grade 1	Grade 2	Grade 1	Grade 2	Grade 1
View of Glottis	Excellent	Excellent	Excellent	Excellent	Excellent	Excellent	Excellent	Excellent	Excellent
Success on First Attempt	Yes	Yes	Yes	Yes	Yes	Yes	Yes	Yes	Yes
Intubation Time	58 s	21 s	20 s	42 s	15 s	23 s	20 s	19 s	9 s
Complications	Nil	Nil	Nil	Nil	Nil	Nil	Nil	Nil	Nil
Subjective Satisfaction	Excellent	Excellent	Excellent	Excellent	Excellent	Excellent	Excellent	Excellent	Excellent

Note: M/F—male/female; BMI—body mass index (kg/m^2^); MDZ—midazolam; FEN—fentanyl; PPF—propofol; ROC—rocuronium; succinylcholine—SCC; intubation time—from lip to trachea; LQS grading of the epiglottis is assessed during the jaw thrust maneuver: Grade 1—the glottic visualization is equivalent to the percentage of glottic opening (POGO) scale from 1% to 100%; Grade 2—POGO scale is 0, but there is still enough space between the epiglottis and the posterior pharyngeal wall. It is not difficult to advance the stylet–endotracheal tube set beneath the epiglottis to reach the glottis; Grade 3—there is marginal space between the epiglottis and posterior pharyngeal wall, and it is expected to be difficult to maneuver the stylet to pass beneath the epiglottis; HIVD—herniation of an intervertebral disc; ACDF—anterior cervical discectomy and fusion; DBS—deep brain stimulation; complications—dental/soft tissue injury, hypoxia, hypertension, arrhythmias, etc.

## Data Availability

Not applicable.

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
