# Peer review of "An Approach to Improve the Effectiveness of the Video-Assisted Intubating Stylet Technique for Tracheal Intubation: A Case Series Report"

_healthcare, 2023, doi:10.3390/healthcare11060891_

Round 1

Reviewer 1 Report (Previous Reviewer 4)

The authors have addressed the comments the reviewer provided and the manuscript has improved substantially.

Author Response

Dear Reviewer

We appreciate your time and effort to give excellent and constructive comments on our manuscript.

Reviewer 2 Report (Previous Reviewer 3)

Dear Authors,

In my opinion, this manuscript is a well-written series of nine cases that provides a comprehensive overview of the video-assisted intubating stylet technique for tracheal intubation.

Thank you for submitting this manuscript.

Best regards,

Author Response

Dear reviewer

We appreciate your time and effort to give the excellent opinions and constructive comments on our manusccript.

This manuscript is a resubmission of an earlier submission. The following is a list of the peer review reports and author responses from that submission.

Round 1

Reviewer 1 Report

Point 1: Title of the manuscript not fulfilled. The technique is mentioned but for what application?

Point 2: In the article, the discussion of COVID-19 is NOT involved. It is suggested to delete the keyword “COVID-19”.

Point 3: In the abstract, it is mentioned that ‘techniques to improve efficiency and accuracy’.     But it is rarely discussed in the content.

Point 4: Highlight how efficiency and accuracy are improved by using an intubating stylet.      

Point 5: Proof for case studies may be updated with proper reference citations.

Point 6: It is suggested that a comparative statement or table about the case studies taken for analysis may be included.

Author Response

Reviewer-1

Comment-1: Title of the manuscript not fulfilled. The technique is mentioned but for what application?

Response-1: Thanks for your comment. The title has therefore been revised as follows:

A makeshift to empower the effectiveness of video-assisted intubating stylet technique for tracheal intubation: A case series report

Comment-2: In the article, the discussion of COVID-19 is NOT involved. It is suggested to delete the keyword “COVID-19”.

Response-2: Thanks for your comment. The keyword of “COVID-19” has therefore been deleted from the Keywords section.

Comment-3: In the abstract, it is mentioned that ‘techniques to improve efficiency and accuracy’.  But it is rarely discussed in the content.

Response-3: Thanks for your excellent comment.

  • In the original text, we have mentioned the common pitfalls of the VS technique in the Introduction section (also see Figures 2 and 3), i.e., the interference of visualization by patient’s secretions, blood, mucus, etc. Such impediment to smooth and swift tracheal intubation should be well taken care of at the outset of airway management.
  • Per your suggestion, we therefore added a new table (Table 1) to list all the relevant details about each case. In this Table 1, we particularly include 5 parameters: (1) Glottic visualization; (2) First-pass success; (3) Intubation time; (4) Complications; (5) Subjective satisfaction. All these indices can be used as an evaluation of whether such makeshift of suction assembly improves the efficiency and accuracy of VS technique for tracheal intubation.

Comment-4: Highlight how efficiency and accuracy are improved by using an intubating stylet.     

Response-4: Thanks for your comment.

  • The efficiency and accuracy of the VS technique itself has been described in the Introduction section (Line 59).
  • We have also referred such notion to our newly published review article [9]. In particular, we would like to emphasize that the VS technique makes tracheal intubation efficient and accurate for both normal/difficult airway and routine/emergent scenarios. This notion will be soon enhanced in our preparing review article to be submitted.
  • We also added a new table (Table 2) to exemplify the advantages of application of VS in comparison to other intubating modalities.

Comment-5: Proof for case studies may be updated with proper reference citations.

Response-5: Thanks for your comment. References citation will be carefully checked and corrected during galley proof.

Comment-6: It is suggested that a comparative statement or table about the case studies taken for analysis may be included.

Response-6: Thanks for your constructive comment. A new table (Table 1) has therefore been added.

Table 1. Summary of the 9 cases

Case 1

Case 2

Case 3

Case 4

Case 5

Age/gender

56/M

58/M

25/M

27/F

47/M

BMI

23.4

36.4

44.7

42.2

23.8

Diagnosis

Cervical HIVD

Parkinson’s disease

Exotropia

Pituitary cyst

Thyroid goiter

Surgery

ACDF

DBS

Recession and myectomy

Transsphenoidal removal

Thyroidectomy

Induction

Midazolam

Fentanyl

Propofol

Recouronium

Fentanyl

Propofol

Rocuronium

Fentanyl

Propofol

Rocuronium

Midazolam

Fentanyl

Propofol

Rocuronium

Fentanyl

Propofol

Rocuronium

Maintenance

Desflurane

Sevoflurane

Sevoflurane

Sevoflurane

Sevoflurane

Airway evaluation

Cervical collar

Stereotactic frame

Enlarged tonsils

LQS grading

Grade 2

Grade 2

Grade 2

Grade 2

Grade 1

Glottic visualization

Excellent

Excellent

Excellent

Excellent

Excellent

First-pass success

Yes

Yes

Yes

Yes

Yes

Intubation time

58 s

21 s

20 s

42 s

15 s

Complications

Nil

Nil

Nil

Nil

Nil

Subjective satisfaction

Excellent

Excellent

Excellent

Excellent

Excellent

M/F: male/female; BMI: body mass index (kg/m2); intubation time: from lip to trachea; LQS grading: grade 1: any part of vocal cords visible, grade 2: no visible vocal cords but there is enough space beneath epiglottis to pass the stylet, grade 3: minimal space between epiglottis and posterior pharyngeal wall and difficult to maneuvering the stylet to pass.

Table 1. Summary of the 9 cases (continued)

Case 6

Case 7

Case 8

Case 9

Age/gender

63/F

45/M

68/F

18/F

BMI

25.6

27.6

30.8

23.4

Diagnosis

Gum tumor

Facial trauma

Deep neck infection

Acute appendicitis

Surgery

Wide excision maxillectomy

Open reduction, graft, wound deb ridement

Incision, drainage andtracheostomy

Laparoscopic appendectomy

Induction

Fentanyl

Propofol

Recouronium

Succinlycholine

Fentanyl

Propofol

Rocuronium

Succinlycholine

MidazolamFentanyl

Propofol

Succinylcholine

Fentanyl

Propofol

Rocuronium

Succinlycholine

Maintenance

Sevoflurane

Sevoflurane

Sevoflurane

Sevoflurane

Airway evaluation

LQS grading

Grade 2

Grade 1

Grade 2

Grade 1

Glottic visualization

Excellent

Excellent

Excellent

Excellent

First-pass success

Yes

Yes

Yes

Yes

Intubation time

23 s

20 s

19 s

9 s

Complications

Nil

Nil

Nil

Nil

Subjective satisfaction

Excellent

Excellent

Excellent

Excellent

M/F: male/female; BMI: body mass index (kg/m2); intubation time: from lip to trachea; LQS grading: grade 1: any part of vocal cords visible, grade 2: no visible vocal cords but there is enough space beneath epiglottis to pass the stylet, grade 3: minimal space between epiglottis and posterior pharyngeal wall and difficult to maneuvering the stylet to pass.

Reviewer 2 Report

Dear Editor

Thanks for your confidence

I think this a very interesting paper that could be useful for the readers.

They should to include a section about limitations of this study (eg unicenter study, a few participating researches, ...).

Authors should change the sentence "Although DL and VL still play important roles in airway management in many countries and regions, the new emergence of the VS technique gradually prevails in recent years." for an other no so favourable to the VS.  Nowadays, VL are usually  used to perform a tracheal intubation. Probably the VS could be useful to facilitate the tracheal intubation in the future for normal o difficult airway patients.

Best regards,

Professor Manuel Granell

Author Response

Reviewer-2

Point-1: They should to include a section about limitations of this study (e.g.; unicenter study, a few participating researchers, ...).

Response-1: Thanks for your comment. A paragraph regarding limitations of the case report has been added as follows.

 “The main limitations of this case series report include (1) the retrospective study design nature; (2) the limited possibility of generalizing the validity and reliability of the study due to the small number of the subjects; (3) the study was conducted by a single airway operator in one single medical center. Therefore, the use and interpretation of case series report need to be carefully considered within a much broader context, in order to avoid unnecessary overstatement of the validity of the study.”

Comment-2: Authors should change the sentence "Although DL and VL still play important roles in airway management in many countries and regions, the new emergence of the VS technique gradually prevails in recent years." for another no so favourable to the VS. “Nowadays, VL are usually used to perform a tracheal intubation. Probably the VS could be useful to facilitate the tracheal intubation in the future for normal o difficult airway patients.”

Response-2: Thanks for your comment. Per your comment, the following sentence replaced the original one.

“Nowadays, VL are usually used to perform a tracheal intubation. Probably the VS could be useful to facilitate the tracheal intubation in the future for normal or difficult airway patients.”

Reviewer 3 Report

Dear Authors,

  you write "Several new tools and techniques have been developed", maybe you can describe the newest patents that made a breakthrough?  Another question is about costs of VS / soft flexible fiberoptic scope?

Author Response

Reviewer-3

Comment-1: You write "Several new tools and techniques have been developed", maybe you can describe the newest patents that made a breakthrough?  Another question is about costs of VS /soft flexible fiberoptic scope?

Response-1: Thanks for your comment.

(1) Regarding the newly developed airway management tools in the last decades, few examples have been added in the text:  “(e.g., videolaryngoscopes, laryngeal masks, optic stylets, etc.)”

(2) The cost of VS is from 300 USD to 5000 USD. The cost of FOB is in the range of 20000 to 30000 USD. The cost of maintenance is different too.

Reviewer 4 Report

The authors describe a technique to improve visualisation during endotracheal intubation by video stylet (VS).

As endotracheal intubation by VS may be hampered by obstruction or secretions the topic is of interest to the specialty. However, this report has limitations that limit the quality of the report and the conclusions which can be appropriately drawn from it.

The following comments are offered for the authors considerations:

Major Considerations:
As the authors indicate, video-assisted techniques provide some advantages over direct techniques. To equate VS with Videolaryngoscopes is in my opinion not adequate and should be discussed thoroughly.

Moreover, the cases you describe are predominantly patients with anticipated difficult airways. The recommended approach with an anticipated difficult airway is to perform awake intubation. Please refer to that topic in your case presentation and discussion.

.

Abstract:

L 19: To my knowledge, there are no valid data refering to intubation time with VS compared to direct layngoscopy. Please specify this further.

Please specify in the abstract the main goal of the report: to offer a technique to improve the visibility of the glottis by introducing a combination of a nasopharyngeal airway with a suctioning catheter.

Introduction:

L 35: Please provide a reference for the shorter intubation time by VL. At least, the time required for tracheal intubation was not different in the Cochrane review of 2022 by Hansel you referenced.

In your introduction your main focus seems to be an introduction to VS. I would recommend to focus more on the difficulties one experiences with the application of VS. You therefore may expand the last paragraph oft he introduction.

The capture of Fig.1 shows different VS but the scope of your paper is the suction tool, please emphasize on this.

Minor comment: the abbreviation VS is not introduced.

Cases:

Please describe in more detail the assembly of the modified suction system/catheter-nasopharyngeal airway assembly Why is it secured with tape? How long the suction catheter is protruding out of the nasopharyngeal airway? Do you introduce the nasopharyngeal tube parallel to the VS or in advance?  

Case 1: Why did you not choose an awake technique?

L 62: Omit „ is“

Figure 4: Please insert the desigbnations A-H within the pictures, like in your other figures.

Case 2: Why did youperform the induction of general anesthesia after the application of the stereotactical headframe?

Figure 6: Please rephrase (A)….needed two assisted…

Case 4: Why did you induce neuromuscular block with a combination of Rocuronium and Suxamethonium?

Moreover, the description of anesthesia, medications, monitoring and outcome is very much copy paste, maybe you can modify these redundancy in all the cases you describe.

L 196: Please omit the „s“ oft he goiter, usually there is only one. Same with the capture of Fig 8.

Case 6: Why did you induce neuromuscular block with a combination of Rocuronium and Suxamethonium?

Case 7: The apply nasal high-flow in a patient with midface maxillofacial trauma is dangerous and commonly a contraindication. Please explain why you choose this approach.

Case 9: Again a combination of Rocuronium and Suxamethonium. Do you still use cricoid pressure? There is controversial discussion about CP during RSII. Please explain why you released CP before starting tracheal intubation.

As a general remark, most of your cases would be managed by awake intubation. Please discuss that.

Discussion:

I would recommend to discuss what status VS in the management oft he difficult airway has. You may provide a classification and recommendation for the reader, when and how a VS ist he preferred choice for endotracheal intubation. Moreover you may describe and clarify the options to remove secretions and improve visibility and make the point to your catheter-nasopharyngeal airway assembly.

An additional topic may be a combined approach (VL with VS), like VL with flexible Scope.

Author Response

Reviewer-4

General Comments: The authors describe a technique to improve visualisation during endotracheal intubation by video stylet (VS). As endotracheal intubation by VS may be hampered by obstruction or secretions the topic is of interest to the specialty. However, this report has limitations that limit the quality of the report and the conclusions which can be appropriately drawn from it. The following comments are offered for the authors considerations:

Comment-1: As the authors indicate, video-assisted techniques provide some advantages over direct techniques. To equate VS with Videolaryngoscopes is in my opinion not adequate and should be discussed thoroughly.

Response-1: Thanks for your excellent comment. We understand the important role of VL for airway management in the past two decades. In order to share different views and clinical experiences of using VS, we added a new table (Table 2) to compare the DL, VL, FOB, and VS during routine and difficult airway management.

Table 2. Comparison of DL, VL, FOB, and VS for airway management: generalized advantages and disadvantages

DL

VL

FOB

VS

Affordability

cheap

expensive

expensive

moderate

Availability

+++

++

+

++

Accessibility

+++

++

+

++

Maintenance

easy

easy

complicated

easy

Learning curve

slow

fast

slower

faster

Team performance

+

+++

++

+++

Rescue for difficult/failed intubation

-

+++

+

+++

Combined use

-

-

with DL/DL/SGA

with DL/VL/FOB

Use in ER, ICU, EMT

+++

+++

+

+++

View on video monitor

-

+++

+++

+++

Mouth wide opening

+++

++

-

+

Glottic visualization

+

++

+++

+++

Lifting force

+++

++

+

-

Need to align airway axes

+++

++

-

-

Need of external manipulation

+++

++

+

-

Difficulty inserting ET tube

++

+

+

-

Difficulty railroading ET tube

-

-

+++

-

Stylet requirement

+++

++

-

-

Impact of collapsed epiglottis

+

+

+++

+++

Impact of secretions on the lens

-

++

+++

+++

First-attempt success rate

+

+++

++

+++

Time to intubate

long

short

long

shorter

Look around the corner

+

++

++

+++

Esophageal intubation

+++

+

+

-

Simulation

+++

++

-

-

Complications

+++

+

+

-

Awake intubation

++

++

+++

++

Subjective satisfaction

+

++

++

+++

Comment-2: Moreover, the cases you describe are predominantly patients with anticipated difficult airways. The recommended approach with an anticipated difficult airway is to perform awake intubation. Please refer to that topic in your case presentation and discussion.

Response-2: Thanks for bringing up this important issue. Indeed, the role of awake intubation for anticipated difficult airway (e.g., expected difficult mask ventilation/rapid desaturation/aspiration/difficult surgical airway) has been emphasized in the new ASA guideline. None of the above four scenarios in general were applied to all of our cases. Nevertheless, we added a new sentence regarding this warning in the Discussion section.

“However, it should be mentioned that, when difficult airway is anticipated/suspected, awake intubation option should be seriously considered and prepared [30].”

  • Apfelbaum et al., American Society of Anesthesiologists Practice Guidelines for Management of the Difficult Airway. Anesthesiology 2022, 136, 31–81. https://doi.org/10.1097/ALN.0000000000004002

Comment-3: Abstract: L 19: To my knowledge, there are no valid data referring to intubation time with VS compared to direct laryngoscopy. Please specify this further.

Response-3: Thanks for your excellent comment. Indeed, it is not easy to compare the intubation qualities of VS with VL/DL in the real world. However, as in the following mannequin study, the intubation time (25.4 s) was shorter using the Aram Stylet than that using the McGrath® video laryngoscope (35.1 s) (p < 0.001).

  • Park JW et al., Comparison of a New Video Intubation Stylet and McGrath® MAC Video Laryngoscope for Intubation in an Airway Manikin with Normal Airway and Cervical Spine Immobilization Scenarios by Novice Personnel: A Randomized Crossover Study. Biomed Res Int. 2021 Nov 10;2021:4288367. doi: 10.1155/2021/4288367.

Comment-4: Please specify in the abstract the main goal of the report: to offer a technique to improve the visibility of the glottis by introducing a combination of a nasopharyngeal airway with a suctioning catheter.

Response-4: Thanks for your comment. The following sentence has been added in the Abstract section.

“…, we describe a technique to improve the visibility of the glottis by introducing a combination of a nasopharyngeal airway with a suctioning catheter and to enhance the efficiency and accuracy in using the intubating stylet for tracheal intubation.”

Comment-5: Introduction: L 35: Please provide a reference for the shorter intubation time by VL. At least, the time required for tracheal intubation was not different in the Cochrane review of 2022 by Hansel you referenced.

Response-5: Thanks for your excellent comment on this issue with nuance. Indeed, due to the extremely high level of statistical heterogeneity, it is difficult to analyze such data in several published meta-analysis articles. In literature, several conditions and scenarios affected the study outcomes and led to inconsistent or even contradictory results and conclusion (e.g., higher first-pass success rate, faster intubation time, etc). We here gave four such examples as follows.

  • Nouruzi-Sedeh P, Schumann M, Groeben H. Laryngoscopy via Macintosh blade versus GlideScope: success rate and time for endotracheal intubation in untrained medical personnel. Anesthesiology. 2009 Jan;110(1):32-7. doi: 10.1097/ALN.0b013e318190b6a7.
  • Ayoub CM, Kanazi GE, Al Alami A, Rameh C, El-Khatib MF. Tracheal intubation following training with the GlideScope compared to direct laryngoscopy. Anaesthesia. 2010 Jul;65(7):674-8. doi: 10.1111/j.1365-2044.2010.06335.x. Epub 2010 May 17.
  • Aziz MF, Dillman D, Fu R, Brambrink AM. Comparative effectiveness of the C-MAC video laryngoscope versus direct laryngoscopy in the setting of the predicted difficult airway. Anesthesiology. 2012 Mar;116(3):629-36. doi: 10.1097/ALN.0b013e318246ea34.
  • BektaÅŸ H, Göksu S, Åžen E. A Comparison of the Effectiveness of Videolaryngoscopy and Macintosh Laryngoscopy in Intubation Attempts on Adult Patients. Turk J Anaesthesiol Reanim. 2022 Oct;50(5):352-357. doi: 10.5152/TJAR.2022.21367.

Aziz et al. (2012) reported use of video laryngoscopy resulting in more successful intubations on first attempt (93%) as compared with direct laryngoscopy (84%). However, laryngoscopy time averaged 46 s for the C-MAC group and was shorter in the direct laryngoscopy group (33 s). In contrast, Nouruzi-Sedeh et al. (2009) reported that the overall success rate was 93% for the GlideScope technique versus 51% for direct laryngoscopy. Meanwhile, the time for intubation was 89 s for direct laryngoscopy versus 63 s for GlideScope technique. Here, we simply quoted the article by Nouruzi-Sedeh et al. (2009) as the new reference-4.

Comment-6: In your introduction your main focus seems to be an introduction to VS. I would recommend to focus more on the difficulties one experiences with the application of VS. You therefore may expand the paragraph of the introduction.

Response-6: Excellent suggestion. We have expanded such issue in the Introduction section.

“Similar to Bonfils intubation endoscope and fiberoptic scopes [23,24], the VS may be encumbered by copious and tenacious secretions, blood or vomitus in the airway (Figure 3). The secretions and mucus then have to be carefully removed by suctioning to improve the visualization of airway anatomy, and the lens of the VS may need to be cleaned off. Another difficulty for VS technique is loss of orientation and course of direction, caused by the patient’s airway structures an anatomy.”

Comment-7: The capture of Fig.1 shows different VS but the scope of your paper is the suction tool, please emphasize on this.

Response-7: Thanks for your note. Indeed, the purpose of the Figure 1 A-D is to make an introduction of the VS to the readers who might not be familiar with. Together with the application of VS, we later focus on the effects of the suction assembly. The captions for the Figure 1F “(F) combined use of VS technique and suction-guide assembly in a mannequin model.” serve the purpose.

Comment-8: Minor comment: the abbreviation VS is not introduced.

Response-8: Corrected as follows! Thanks.

  “… this video-assisted intubating stylet (VS, the Shikani technique)”

Comment-9: Cases: Please describe in more detail the assembly of the modified suction system/catheter-nasopharyngeal airway assembly. Why is it secured with tape? How long the suction catheter is protruding out of the nasopharyngeal airway? Do you introduce the nasopharyngeal tube parallel to the VS or in advance? 

Response-9: Great suggestion. To illustrate the device and the intubating technique, we therefore added a new figure (Figure 4) to describe the suction assembly in more detail.

  • To tape the suction catheter with the nasopharyngeal airway is to secure the tip of the former in a proper length and position.
  • The length of the suction catheter tip protruding out of the nasopharyngeal airway is about 2 cm.
  • We always introduce the nasopharyngeal airway into patient’s mouth before applying the VS. The purpose of such suction assembly is to clear any possible secretions to avoid obscuring the lens of the VS.

Comment-10: Case 1: Why did you not choose an awake technique?

Response-10: Great question. We understand the option of “awake technique” for airway management, especially for those scenarios as anticipated/suspected difficult airway. With our vast clinical experiences of using VS technique, we are prone to apply the VS technique, e.g., in patients with limited cervical spine mobility. We would like to refer our experience and view on this issue to the reference 21.

Comment-11: L 62: Omit „ is“

Response-11: Have omitted “is” and revised the original sentence as “…, the VS may be encumbered by…”

Comment-12: Figure 4: Please insert the designations A-H within the pictures, like in your other figures.

Response-12: The Figures (the new Figures 5 and 8) have been revised accordingly.

Comment-13: Case 2: Why did you perform the induction of general anesthesia after the application of the stereotactical headframe?

Response-13: Accuracy of stereotactic coordinate transformation using a Leksell G frame (Elekta Instrument Inc., Norcross, GA, USA) and computed tomographic imaging (and MRI) for stereotactic deep brain stimulation procedure has been developed for decades. In the past, our frame-based stereotactic localization was indeed processed as follows: first to anesthetize and intubate the patients, mount the stereotactic headframe and affix to the patient’s head, then send the patient them to the CT room to complete the localization of the targets, and finally transport the patient back to the operating room for surgery. Nowadays, because the revision of the design of the headframe and improvement of the tracheal intubation technique, we routinely anesthetized and intubated the patients after the headframe was mounted and stereotactic localization and labeling being accomplished (please see the following references).

  • Chen SY et al., Subthalamic deep brain stimulation in Parkinson's disease under different anesthetic modalities: a comparative cohort study. Stereotact Funct Neurosurg. 2011, 89, 372-380. doi: 10.1159/000332058.
  • Tsai ST et al., Five-Year Clinical Outcomes of Local versus General Anesthesia Deep Brain Stimulation for Parkinson's Disease. Parkinsons Dis. 2019 Jan 17;2019:5676345. doi: 10.1155/2019/5676345.
  • Chen YC et al., Median Nerve Stimulation Facilitates the Identification of Somatotopy of the Subthalamic Nucleus in Parkinson's Disease Patients under Inhalational Anesthesia. Biomedicines. 2021 Dec 30;10(1):74. doi: 10.3390/biomedicines10010074.

Comment-14: Figure 6: Please rephrase (A)….needed two assisted…

Response-14: As indicated, the legend of the Figure 6 has been rephrased.

“…. and needed an assistant to help to open the patient’s airway”

Comment-15: Case 4: Why did you induce neuromuscular block with a combination of Rocuronium and Suxamethonium?

Response-15: Excellent point. High dose of rocuronium does achieve the rapid onset of neuromuscular block for emergent tracheal intubation (e.g., rapid sequence induction and intubation) and could be readily chelated by sugammadex in case failed intubation occurred. In Taiwan, unfortunately, patients need to pay for sugammadex at their own expense (i.e., the universal health insurance does not cover this medication). Therefore, both rocuronium and succinylcholine are optional on daily anesthesia practice. When succinylcholine is used for tracheal intubation, prior use of small dose of rocuronium (or other non-depolarizing NMBAs) is to alleviate the side effects of succinylcholine (e.g., fasciculations), and perhaps to enhance the muscle relaxation by succinylcholine.  

Comment-16: Moreover, the description of anesthesia, medications, monitoring and outcome is very much copy paste, maybe you can modify these redundancy in all the cases you describe.

Response-16: It has been our plan not to repeat the description and statement for each case in the case report section. However, usually we were required to comply with the guidelines (e.g., CARE case report guidelines) and to specify the diagnoses, surgery, monitoring, medication, and outcomes etc for each case. Therefore, we apologize the redundancy of the text in the case report section. We added a new table (Table 1) to summarize all the cases for the readers’ convenience.

Comment-17: L 196: Please omit the “s“ of the goiter, usually there is only one. Same with the capture of Fig 8.

Response-17: Good point. It has be revised as “goiter” without the “s” in both the text and figure cations accordingly.

Comment-18: Case 6: Why did you induce neuromuscular block with a combination of Rocuronium and Suxamethonium?

Response-18: Same issue as the comment-15. Please refer to our response-15.

Comment-19: Case 7: The apply nasal high-flow in a patient with midface maxillofacial trauma is dangerous and commonly a contraindication. Please explain why you choose this approach.

Response-19: Great comment. We apologize the misunderstanding we caused here. Our original text is as follows:  “Pre-oxygenation with high-flow O2 was given...” Here we mean the face mask ventilation with high flow O2 (10 L/min) for pre-oxygenation. It is not the nasal high-flow nasal cannula (HFNC) which we usually applied in patients with morbid obesity, COVID-19, or for tubeless general anesthesia.  Indeed, maxillofacial trauma is one of the contraindications to use HFNC and other non-invasive ventilation modalities. The following sentence has been added to clarify this issue.

“Pre-oxygenation with high-flow O2 via face mask ventilation was given.”

Comment-20: Case 9: Again a combination of Rocuronium and Suxamethonium. Do you still use cricoid pressure? There is controversial discussion about CP during RSII. Please explain why you released CP before starting tracheal intubation.

Response-20:

(1) Same issue as the comment-15. Please refer to our response-15.

(2) Great comment on releasing CP before intubation. Indeed, the role of Sellick maneuver (cricoid pressure, CP) for RSI is controversial and currently under debate (please see the following references).

  • Snider et al., The "BURP" maneuver worsens the glottic view when applied in combination with cricoid pressure. Can J Anaesth. 2005, 52, 100-104. doi: 10.1007/BF03018589.
  • Corda DM, Riutort KT, Leone AJ, Qureshi MK, Heckman MG, Brull SJ. Effect of jaw thrust and cricoid pressure maneuvers on glottic visualization during GlideScope videolaryngoscopy. J Anesth. 2012, 26, 362-368. doi: 10.1007/s00540-012-1339-0.
  • Algie et al., Effectiveness and risks of cricoid pressure during rapid sequence induction for endotracheal intubation. Cochrane Database Syst Rev. 2015, Nov 18;2015(11):CD011656. doi: 10.1002/14651858.CD011656.pub2.
  • Salem et al., Cricoid Pressure Controversies: Narrative Review. Anesthesiology. 2017, 126, 738-752. doi: 10.1097/ALN.0000000000001489.
  • Lin YC, Cho AH, Lin JR, Chung YT. The Clarus Video System (Trachway) and direct laryngoscope for endotracheal intubation with cricoid pressure in simulated rapid sequence induction intubation: a prospective randomized controlled trial. BMC Anesthesiol. 2019 Mar 4;19(1):33. doi: 10.1186/s12871-019-0703-0.
  • Won et al., Effect of Paratracheal Pressure on the Glottic View During Direct Laryngoscopy: A Randomized Double-Blind, Noninferiority Trial. Anesth Analg. 2021, 133, 491-499. doi: 10.1213/ANE.0000000000005620.
  • Zdravkovic et al., The Clinical Use of Cricoid Pressure: First, Do No Harm. Anesth Analg. 2021, 132, 261-267. doi: 10.1213/ANE.0000000000004360.
  • Kim H, Chang JE, Won D, Lee JM, Kim TK, Kim MJ, Min SW, Hwang JY. Effectiveness of Cricoid and Paratracheal Pressures in Occluding the Upper Esophagus Through Induction of Anesthesia and Videolaryngoscopy: A Randomized, Crossover Study. Anesth Analg. 2022, 135, 1064-1072. doi: 10.1213/ANE.0000000000006154.
  • Both Sellick maneuver and BURP might be beneficial to laryngoscopy technique. However, such maneuvers might not be helpful to VS technique. One of the drawbacks of such maneuvering is to compress and distort the glottis, causing longer time to localize the glottis and more difficult to railroad the stylet-ET tube unit.
  • In our daily routine practice with VS technique, most of our colleagues still applied cricoid pressure for RSII. In contrast, I personally prefer to adopt Sellick maneuver during pre-oxygenation phase. But immediately after insertion of the VS into patient’s oropharyngeal space, the cricoid pressure by the assistant is about to relieve. The visualization of the glottis under the VS is always satisfactory.

Comment-21: As a general remark, most of your cases would be managed by awake intubation. Please discuss that.

Response-21:  Indeed, when anticipated or predicted difficult airway was recognized, awake intubation is the option, according to the recent published ASA guideline (Apfelbaum et al., 2022).

  • Apfelbaum et al., 2022 American Society of Anesthesiologists Practice Guidelines for Management of the Difficult Airway. Anesthesiology. 2022, 136, 31-81. doi: 10.1097/ALN.0000000000004002.

However, if the patient is not contraindicated for anesthetized intubation and meanwhile the airway operator (and the team) feel comfortable and competent (and with well-prepared plan B), we believe the anesthetized intubation is also optional. Just like the cases we presented in the article (Figures 5-13), with VS technique, the outcomes were excellent. Per your suggestion, we added the following sentence in the Discussion section.

“However, it should be mentioned that, when difficult airway is anticipated/suspected, awake intubation option should be seriously considered and prepared [30].”

Comment-22: Discussion: I would recommend to discuss what status VS in the management of the difficult airway has. You may provide a classification and recommendation for the reader, when and how a VS is the preferred choice for endotracheal intubation.

Response-22: A great suggestion.

It is indeed our intent to propose an algorithm for the role of VS in routine/ emergency, and anticipated/unexpected difficult airway management. However, this issue is big and beyond the scope of this brief case report. Actually, we are preparing a manuscript to address this issue of algorithm regarding VS.

Comment-23: Moreover you may describe and clarify the options to remove secretions and improve visibility and make the point to your catheter-nasopharyngeal airway assembly.

Response-23: A great suggestion. We have added a statement regarding this issue.

“During regular anesthesia induction and tracheal intubation process, both the regular flexible suction catheter and a rigid Yankauer suction are commonly used for adequate suction to remove any substances (saliva, fluid, mucus, blood, secretions, vomitus, etc.) which might impede the visibility of the airway. The speed and adequacy of the ability of suction technique to remove the substances from the glottis area assure the rapid and smooth tracheal intubation. In contrast to SALAD technique, our soft suction catheter-flexible nasopharyngeal airway assembly can also effectively eliminate fair amount of the secretions and mucus during routine tracheal intubation procedure (as demonstrated in Figures 5-13). ”

Comment-24: An additional topic may be a combined approach (VL with VS), like VL with flexible Scope.

Response-24: A great suggestion. We have added this issue in the Discussion section. Actually, we just published the role of combining VL and VS for airway management (reference 22). The follow sentence has been added in the Discussion section.

“It is worthy to mention that the combined used of VS with VL/DL [22] in patients with copious secretions in the airway or with a large floppy could be useful.”

Round 2

Reviewer 4 Report

The authors have addressed most of the comments the reviewer provided and the manuscript has improved substantially.

There are still some minor limitations I suggest to improve the quality of the report.

Abstract:

L 19: The reference (Park JW et al) concerning intubation time with VS compared to direct laryngoscopy is manikin study. You should clarify that in the abstract and in the introduction. Moreover, it is questionable that an unskilled provider is able to be faster with a tool he uses seldom. A well-experienced airway lead may be faster with the VS, unfortunately this is not the reality of clinical anesthesia.  

Introduction:

L 42: The references for the shorter intubation time by VL show conflicting results. The referenced article by Nouruzi-Sedeh et al used untrained personel (untrained in DL and VL). This does not reflect clinical reality. Anesthesia providers are usually trained and experienced with DL. In my opinion it is not scientific sound to state “shorter intubation time” with VL versus DL. You may at least discuss that in the discussion section.

Minor comment: L 86 “an” may be better “and”

Cases:

Case 2: I would still like to question your approach of intubating the patient after the stereotactical headframe is mounted. If a tracheal intubation is planned anyway, why do you mount the headframe with local infiltration and expose the fully awake patient to the stress of a mounted frame, where every even small vibration causes maximal distress?
In my institution we also care for “awake” patients with a mounted stereotactical headframe, but only if an awake patient is mandatory for the success of the stereotactical procedure (e.g. target confirmation). The three references you cite to underline your approach is not underlining your approach. Please correct this.

Case 4: Why did you induce neuromuscular block with a combination of Rocuronium and Suxamethonium?

You explain in your response-15 that you use Rocuronium as a “priming” drug, a small dose of NDMR before the application of Succinylcholine. You may just describe that as a priming dose of ROC followed by a relaxation dose of SUX.

Case 9: There is indeed ongoing controversial discussion about CP during RSII. You may know that a release of CP before securing the airway may even increase the risk of aspiration because CP leads to an insufficient lower esophageal sphincter tone and a regurgitation of gastric content just up to the upper esophageal area where the CP hopefully inhibit the fluid to go straight to the hypopharyngeal space and the larynx.

In my opinion it is more dangerous to relieve CP before tracheal intubation is accomplished instead of applying no CP. Please discuss that in the discussion section.

Response-21

I would strengthen your statement “However, it should be mentioned that, when difficult airway is anticipated/suspected, awake intubation option should be seriously considered and prepared [30].” by “According to current guidelines”

Author Response

Reviewer 4

20220216

The authors have addressed most of the comments the reviewer provided and the manuscript has improved substantially.

There are still some minor limitations I suggest to improve the quality of the report.

Abstract:

Comment-1: L 19: The reference (Park JW et al) concerning intubation time with VS compared to direct laryngoscopy is manikin study. You should clarify that in the abstract and in the introduction. Moreover, it is questionable that an unskilled provider is able to be faster with a tool he uses seldom. A well-experienced airway lead may be faster with the VS, unfortunately this is not the reality of clinical anesthesia. 

Response-1:

  • Thanks for your excellent point. We understand that the discrepancy and contributing factors in the literature regarding the intubation time among various DL and VL. However, it was not the main purpose for our brief report to make such comparison. Therefore, in the revised text, we simply state in Line 37 “Some other advantages of VL over DL are short intubation time, less sympathetic over-stimulation, and fewer airway injuries [4-6].
  • Regarding the real-world experience on the competency and effectiveness issues among various intubating tools and modalities, we cannot agree with you more. Such comparative clinical studies are difficult to conduct due to many limitations in the real world. We will address this issue in our next manuscript. Here, we would like to cite our related observation (and some others also reported similar findings before) as follows:
  • Ong et al., MH. Comparison between the Trachway video intubating stylet and Macintosh laryngoscope in four simulated difficult tracheal intubations: A manikin study. Ci Ji Yi Xue Za Zhi. 2016 Jul-Sep;28(3):109-112. doi: 10.1016/j.tcmj.2016.06.004. “The TTI was significantly shorter in both the tongue edema and combined scenarios with the TVI stylet compared with the Macintosh laryngoscope (21.60 ± 1.45 seconds vs. 24.07 ± 1.58 seconds and 23.73 ± 2.05 seconds vs. 26.6 ± 2.77 seconds, respectively). The learning time for tracheal intubation using the TVI stylet in difficult airway scenarios was short. Use of the TVI stylet was easier and required a shorter TTI for tracheal intubation in the tongue edema and combined scenarios.”

Introduction:

Comment-2: L 42: The references for the shorter intubation time by VL show conflicting results. The referenced article by Nouruzi-Sedeh et al used untrained personel (untrained in DL and VL). This does not reflect clinical reality. Anesthesia providers are usually trained and experienced with DL. In my opinion it is not scientific sound to state “shorter intubation time” with VL versus DL. You may at least discuss that in the discussion section.

Response-2: Thanks for your point.

  • “Some other advantages of VL over DL are short intubation time,…[4-6].”
  • We understand the real-world issue. However, we believe both trained/experienced and untrained/novice operators are important in the real-world. Therefore, learning and teaching program are important.
  • We agree with you. Therefore, in our revised text, line 20 said “rapid intubation time”, line 37 said “some other advantages of VL over DL are short intubation time,…”. We did not use the word “shorter” because ours is a case series report.

Comment-3: Minor comment: L 86 “an” may be better “and”

Response-3: Corrected. Thanks.

Cases:

Comment-4: Case 2: I would still like to question your approach of intubating the patient after the stereotactical headframe is mounted. If a tracheal intubation is planned anyway, why do you mount the headframe with local infiltration and expose the fully awake patient to the stress of a mounted frame, where every even small vibration causes maximal distress?

In my institution we also care for “awake” patients with a mounted stereotactical headframe, but only if an awake patient is mandatory for the success of the stereotactical procedure (e.g. target confirmation). The three references you cite to underline your approach is not underlining your approach. Please correct this.

Response-4: Thanks for your excellent comment on this important issue.

  • We understand there are several different options that the DBS team can adopt for such neurosurgery. For example, some prefer awake craniotomy and others prefer anesthetized craniotomy. Similarly, some prefer to mount the headframe in advance and stereotactically localize the target in the CT room and then send the patient back to OR for anesthesia and intubation. Either is OK as long as the team works together and improve the clinical performance and patients’ outcomes.
  • Our neurosurgical team for DBS shares the largest volume of such operation in Taiwan (more than 600 cases so far). The reason we cited the three references (doi: 10.1159/000332058; doi: 10.1155/2019/5676345; doi: 10.3390/biomedicines10010074.) in our response letter was to indicate our performance and routine airway management modality (i.e., intubate the patient when headframe has been mounted). Our surgeon colleagues are very satisfied with our airway management on their DBS patients so far.

Comment-5: Case 4: Why did you induce neuromuscular block with a combination of Rocuronium and Suxamethonium? You explain in your response-15 that you use Rocuronium as a “priming” drug, a small dose of NDMR before the application of Succinylcholine. You may just describe that as a priming dose of ROC followed by a relaxation dose of SUX.

Response-5: Excellent suggestion. We have added the statement “A priming dose of rocuronium, followed by a relaxation dose of succinylcholine, was applied to lessen muscular fasciculations and to enhance muscle relaxation for the tracheal intubation.” In the revised text (line 259).

Comment-6: Case 9: There is indeed ongoing controversial discussion about CP during RSII. You may know that a release of CP before securing the airway may even increase the risk of aspiration because CP leads to an insufficient lower esophageal sphincter tone and a regurgitation of gastric content just up to the upper esophageal area where the CP hopefully inhibit the fluid to go straight to the hypopharyngeal space and the larynx. In my opinion it is more dangerous to relieve CP before tracheal intubation is accomplished instead of applying no CP. Please discuss that in the discussion section.

Response-6: An excellent point. Indeed, whether CP provides clinical benefits on preventing/reducing the risks of aspiration during tracheal intubation remains controversial. As we explained previously, this discrepancy in conclusion from literature comes from many contributing factors. We have added the statement “Although the role of CP for RSII remains controversial, we should be cautious that sudden release of CP might be at risk of regurgitation.” for the case 9 (line 342).

Comment-7: Response-21 I would strengthen your statement “However, it should be mentioned that, when difficult airway is anticipated/suspected, awake intubation option should be seriously considered and prepared [30].” by “According to current guidelines

Response-7: A sentence “According to the current guidelines, ..” has been added in the revised text (line 469). 
